# Review of the Effects of Enclosure Complexity and Design on the Behaviour and Physiology of Zoo Animals

**DOI:** 10.3390/ani13081277

**Published:** 2023-04-07

**Authors:** Cristiano Schetini de Azevedo, Cynthia Fernandes Cipreste, Cristiane Schilbach Pizzutto, Robert John Young

**Affiliations:** 1Departamento de Biodiversidade, Evolução e Meio Ambiente, Instituto de Ciências Exatas e Biológicas, Universidade Federal de Ouro Preto, Campus Morro do Cruzeiro, s/n Bauxita, Ouro Preto 35400-000, Brazil; 2Belo Horizonte Zoo, Avenida Otacílio Negrão de Lima, 8000, Pampulha, Belo Horizonte 31365-450, Brazil; 3Programa de Pós-graduação em Reprodução Animal, Faculdade de Medicina Veterinária, Universidade de São Paulo, Avenida Dr. Orlando Marques de Paiva, 87, Cidade Universitária Armando Salles de Oliveira, São Paulo 05508-270, Brazil; 4School of Science, Engineering and Environment, University of Salford Manchester, Peel Building—Room G51, Salford M5 4WT, UK

**Keywords:** captivity, enclosure, habitat complexity, welfare

## Abstract

**Simple Summary:**

Habitat complexity is important for the maintenance of high levels of welfare for captive animals, especially at zoos. Generally, individuals who experience greater enclosure complexity express higher diversity of behaviours and show better physiological well-being. However, positive outcomes of providing habitat complexity should be species-specific, and not all species would benefit from it. Thus, it is important to provide and constantly evaluate the habitat complexity of zoo animals. Complexity can change temporally and spatially. We discuss in this paper how habitat complexity can positively influence animal welfare, and provide some ideas on how to add habitat complexity and functional structures for captive animals.

**Abstract:**

The complexity of the habitat refers to its physical geometry, which includes abiotic and biotic elements. Habitat complexity is important because it allows more species to coexist and, consequently, more interactions to be established among them. The complexity of the habitat links the physical structure of the enclosure to the biological interactions, which occur within its limits. Enclosure complexity should vary temporally, to be able to influence the animals in different ways, depending on the period of the day and season and throughout the year. In the present paper, we discuss how habitat complexity is important, and how it can positively influence the physical and mental states of zoo animals. We show how habitat complexity can ultimately affect educational projects. Finally, we discuss how we can add complexity to enclosures and, thus, make the lives of animals more interesting and functional.

## 1. Habitat Complexity: Definition, Temporal Variation, and Measures

The complexity of a habitat refers to its physical geometry, which includes abiotic components, such as its size, the density and arrangement of structural elements (rocks, waterways, soil, noise, lighting, etc.) and biotic components (e.g., the diversity of plant species and the structures created by them), and the presence of other individuals of the same or different species, for example [1,2,3]. The three-dimensional complexity created by the interaction of biotic and abiotic components is important because it generally allows for the maintenance of greater biodiversity [4]. Therefore, with more species in the habitat, more ecological interactions are recorded [5] and, consequently, the possibility of expression of a greater number of positive behaviours is also increased [6,7].

Animals live in a physical world of three dimensions, a temporal dimension (i.e., time), and a social world. The physical world may be constituted of the biotic and abiotic environment, whereas time is purely abiotic and the social world is biotic [1,2,3]. It is the interaction between these three factors that create the animal’s world and gives it complexity. 

A key question concerns how important complexity is to different animal species, and if it is important for all species. The simplest approach to answering this question would be to assume that species have evolved adaptations to permit them to thrive in their natural environment [8], and that many ecological/biological factors affect a species’ experience of environmental complexity. Examples of these factors are the size of the animal, climatic seasonality, daylength, activity period, migratory or not, habitat use, social complexity, and sensory capabilities. It is important to note that these factors are generalisations that inform a species’ likely experience of environmental complexity, and that they are not mutually exclusive, as many species will fall into extremes of several factors, which may contradict one another.

The size of a species will affect its perception of habitat complexity. For example, a pygmy marmoset (*Cebuella pygmae*; weight = 85–140 g) [9] will perceive the same tree to be far more physically complex than a white bellied spider monkey (*Ateles belzebuth.*; weight = 7800–9400 g) [9]. Perception of complexity is a function of being able to define scale with different levels of accuracy [3]. For example, if you measure the coastline of the U.K. with a measuring tape 1 metre long, this will provide you with a very different length from one that can measure in millimetres. Animals may also perceive temporal scales differently, and so complexity over time might have different meanings for a fly and an elephant [10]. This is discussed further in this paper.

Seasonality, daylength, and the activity period of the day (day or night) would also affect animals’ perception of habitat complexity. Animals living in equatorial regions experience only two distinct seasons (e.g., rainy and dry), whereas animals living in temperate regions experience four distinct seasons [11]. Moreover, in both equatorial and temperate regions, different microclimates can also add variation to animals’ perception of habitat complexity [12,13]. Furthermore, the closer you get to the equator, the less variation there is in daylength, which varies tremendously as you get closer to the polar regions [3]. Finally, habitats can vary in their complexity significantly throughout a 24-h day, and the main activity period of the animal can influence its experience of habitat and social complexity. Given that 70 percent of mammals are nocturnal [14], the opportunities to interact either positively or negatively (e.g., competition, predation, etc.) with other species will vary across this time period. This has significant implications not only for single-housed species (e.g., the need to provide climate needs for animals outside of their normal range) but also for housing different species together in the same exhibit (i.e., mixed-species exhibits).

Another important factor that can influence complexity perception by animals is how complex are their habitats and how they use their habitats. Does the species live mainly in two dimensions (e.g., a Brazilian tapir *Tapirus terrestris*) or three dimensions (e.g., a black-fronted titi monkey *Callicebus nigrifrons*); that is, could a species’ space use best be described as an area or a volume? Larger species are more likely to use an area than a volume, except for marine species [15]. Regarding habitat three-dimensional complexity, a rainforest will be more complex than a savannah, but not all rainforests are equally complex. Rainforests in tropical regions will display greater complexity due to the higher number of plant species [16]. Niche separation in these regions will create not just a greater diversity of species but of species that are physically different [17]. In general, these high-biodiversity areas are found around the equator [18,19]. Regarding the use of the habitat, animal species can use different parts depending on the period of the year and migratory species can experience different habitats during their migrations. For example, polar bears (*Ursus maritimus*) spend time in the tundra and on the Arctic ice during the year [20]. Black bears (*Ursus americanus*) have smaller ranges in comparison to polar bears, but they can thrive in a larger variety of habitats [21,22]. Migrating species have the potential to experience high levels of habitat complexity during their migrations, especially if they migrate slowly or in phases over long distances (north to south or vice versa). On a smaller scale, range-shifting species—that is, emigrating and immigrating animals—could also experience different habitat complexity during the movements between sites.

Social complexity is a function of group size and dynamics and can influence the capacity of a species to perceive complexity [23]. Large and frequently changing groups have the most complexity, such as those shown by species living in fission–fusion societies (e.g., spider monkeys *Ateles* spp., chimpanzees *Pan troglodytes*, muriquis *Brachyteles* spp. and bottlenose dolphins *Tursiops truncatus,* amongst others). However, although species with larger brain sizes normally perceive social complexity [24,25], the ability to perceive and need this complexity is also observed in species with smaller brain sizes (ants, schooling fishes, naked mole rats, etc.) [26,27,28].

Finally, the sensory capabilities of the species will influence their capacity to perceive habitat complexity. Species will vary in their abilities to perceive complexity in different sensory modalities. Birds will perceive greater colour complexity than mammals (i.e., tetrachromats versus large dichromats) [29,30]. Diurnal fruit bat species (Pteropodidae) will perceive habitat complexity differently from a more nocturnal leaf-nosed bat species (Phyllostomidae), since the fruit bats are more visual than leaf-nosed bats, who use more echolocation [31]. This is an important point to consider, especially as animal exhibits are designed to be perceived by the dominant senses of humans (i.e., trichromatic vision and hearing). Thus, animals whose primary senses are sound/echolocation, scent, electric fields, and touch, for example, will perceive habitat complexity differently from humans.

In terms of zoo animal exhibits, it is humans who decide the level of complexity that an animal experiences by co-varying the aforementioned factors. All of these factors should be considered during exhibit planning. Due to construction and planting costs, the level of complexity of an enclosure is a function of its initial design. Logistical constraints mean that moving around key structural components such as walls or roofs is not possible. The same is true for large vegetation such as trees and bushes. Moreover, a balance between habitat complexity for the animals and visitor visibility should be achieved. Thus, when designing an enclosure, complexity needs to be planned considering not only the animal experience but also the visitor and keeper experiences. Additionally, the structures that increase visibility and opportunities for the visitors to observe the animal (e.g., training walls, interactive exhibits, glass windows, media, etc.) need to be associated with structures that may make animals less visible and more stimulated (plants, burrows, shelters, etc.). Exhibits should be complex not only in the exhibit area but also in the animal holding areas (i.e., off-exhibit or bedroom areas), which are often neglected. Thus, enclosure planning must consider the entire facility and not only the areas observed by zoo visitors.

Initially, a zoo exhibit can be perceived to be complex by its inhabitants, but if that complexity is unvarying, then its positive impact on animal welfare will fade with time [32]. To combat this situation, zoos typically will add complexity to their exhibits in the form of environmental enrichment [33]. There are several different categories of environmental enrichment, which themselves may have physical or social complexity, but by frequently changing them, they have temporal complexity (e.g., mirrors that can be used as a physical, social and/or sensorial enrichment, and a puzzle box that can be used as a cognitive and social enrichment [33,34,35,36]). The use of environmental enrichment can offset the lack of complexity built into an exhibit, due to its novelty, but it also can be limited by the design of the available space. For example, exhibit rotation, a type of enrichment consisting of permitting the animals to move between different exhibits, would be impossible to provide if tunnels and bridges are not available in the original design of the enclosures [37].

In a captive environment, the primary role of plants is to create an environment that allows animals to express the same type of movements (i.e., act as a locomotor substrate) and as many natural behaviours as in the wild (for species living in a three-dimensional environment). Moreover, the presence of plants and other structures that make the environment more complex, such as perches and ropes, is important to reduce the boredom caused by barren, low-stimulating enclosures, and by the management routine [38,39,40]. In addition, these structures help animals to cope with the presence of visitors, as they can provide physical barriers against noise produced by the public [41,42], or provide hiding places in case individuals do not want to be visible to visitors [43,44]. Burrows also function as hiding places for the animals, and their existence inside zoo enclosures can enhance habitat complexity. Burrows can be important for protection against negative social interaction, inclement weather, light pollution, and noise pollution, but also for privacy, reproduction, etc. If animals are allowed to choose between different luminosities, temperatures, and noise levels inside an enclosure, they will use the most suitable area for their physical and mental comfort at any given moment [45,46]. However, it is important to remember that if we offer a limited range of choice, animals will select the least adverse one, because options that promote good welfare may not be available. Complex and dynamic physical and social environments, which provide choices and encourage appropriate behavioural responses, provide a sense of control and reduce boredom by making situations less predictable [47].

Furthermore, in a structurally complex enclosure, animals experience different substrates and micro-environments, which are capable of stimulating all or most of the individuals’ sensory organs, allowing complex interactions between them and the environment [48,49]. These complex interactions influence the behaviour displayed by animals, with more behaviours being displayed in more complex enclosures [50,51], and the animals’ physiology, with the activation of physiological cascades that allow the maintenance of homeostasis, decreasing the release of corticosterone and catecholamines, stress-related hormones [52,53,54]. Knowing that animals have at the same time different and specific needs throughout their lives, depending on their sexes, ages, etc. (e.g., young animals like to explore new food sources in comparison to adult conspecifics while older animals may need modifications to allow for restricted movement and other reduced abilities [54]), zoo professionals should plan and design environments which incorporate micro-habitats for the needs of a particular animal. With habitat management, it is possible to meet the specific needs of the species, allowing modifications to respond to individual needs and preferences over time [55]. Thus, designing for complexity (flexible, changeable environments, i.e., not static) allows animals to manage their behaviour, provides them with choice, and allows them to have an experience beyond their basic biological needs.

Another point that should be considered when managing the habitat of zoo animals is that individuals also differ in their needs depending on their personality and biological state. For example, bolder individuals will be more prone to explore the enclosure than shier individuals [56]. Thus, bolder individuals will benefit more if their enclosures offer more complexity because they would use more space, experiencing more micro habitats, substrates, etc. [57,58]. Habitats with more visual barriers, that is, more complex habitats animal personalitycan decrease the number of social interactions [58]. In this way, less sociable individuals would benefit from habitats offering such structures. Such barriers that allow the avoidance of social contact may reduce aggression and competition for resources [32,59]. Some biological states, such as pregnancy and lactation, can influence how females use their habitat. Little brown bats (*Myotis lucifugus*), for example, decrease foraging and home-range size during lactation [60,61]. For captive red pandas (*Ailurus fulgens*), stress-induced behaviours such as pacing were influenced by habitat complexity, with more pacing being expressed when more logs on the ground were present and less pacing being expressed when more tall trees were present in the enclosure [61]. These examples show how habitat complexity can influence the welfare of the animals and how difficult it is to fulfil all an animal’s needs in a captivity.

The complexity of the enclosure can be increased by associating structures that change their three-dimensional configuration to social complexity, that is, individuals of the same or other species [62,63,64,65]. From an enclosure design perceptive, increased social complexity would mean multiple means of feeding opportunities (i.e., resulting in reduced monopolisation), routes of movement through the facility (multiple shift doors, etc.), and varied environments for resting, thermoregulation, and other aspects which may drive competition, socialisation, and individual comfort. From an individual perception, the possibility of intra- and interspecific interactions facilitates the development of individuals’ social skills, allowing the formation of coalitions [65], sexual pairings [66], and hierarchies [67]. Social engagement may be more important for some species than the environmental complexity, enrichment or human–animal interactions, especially in barren exhibits [68]. All these possibilities affect the welfare of animals [69,70], either by reducing the possibility of physical injuries arising from fights between individuals after the definition of hierarchies, or by strengthening ties between individuals [71]. Many species in the wild form polyspecific associations [72,73]. Although mixed-species bird exhibits are present in many zoos [49,74], the number of mixed-species exhibits are only increasing for some groups of animals, such as mammals (one may refer to [75], a site with lists of mixed-species exhibits of carnivores in European zoos). However, the effects of maintaining mixed-species exhibits need to be investigated, since some negative effects can occur such as a decrease in breeding success due to disturbance/competition or frequent changes in composition of species/individuals [76]. Other challenges for mixed-species exhibits are the transmission of diseases, aggression, and nutritional problems due to food competition [77].

The behavioural and physiological effects of a life in a complex captive environment are mostly positive, increasing the animals’ positive experiences and elevating their welfare [78,79]. Creating complex exhibits helps to increase the positive experiences of animals across the five domains of animal welfare [32,59], namely environmental, behavioural interactions, health, and nutritional domains, with a consequent positive outcome on the mental domain [60,61]. This is provided that exhibits enable individual choice and control over the environment. It is important to note that not all negative experiences are bad for well-being; some negative experiences, if not chronic, can lead to a good mental state.

Environmental control and choice are important features to ensure animal welfare in captive environments [33]. An increasing body of empirical evidence is demonstrating that when animals can choose among options inside their enclosures, positive behavioural and physiological changes occur. For example, Amur tigers (*Panthera tigris altaica*) increase their locomotion through different habitats and decrease inactivity when they have access to two different enclosures [80]. When laying hens (*Gallus gallus*) can choose preferable habitats, low blood glucose levels and heterophil:lymphocyte ratios are observed, indicating less stress and better welfare [81]. Several other examples of how choice and control positively affect animal welfare can be found in specific scientific literature [78,82,83]. Thus, complex habitats are important for captive animals, as they offer animals a wide variety of stimuli (choice and control), allowing the expression of normal behaviours [84,85] and the reduction in deleterious effects from the distress sometimes caused by captive life [61,86].

The complexity of an enclosure is not fixed, changing seasonally [1,87]. Differences in solar incidence can modify the luminosity and temperature of the enclosure at different times of the year, which may require the use of heaters and special heating lamps, influencing the selection of micro-habitats by the animals [87,88]. The geographical location, the seasonal variations, the daylight hours and climatic variations and the changing seasons can affect enclosure quality, quantity, and outdoor access [55]. Animals should have free access to outdoor and indoor areas whenever they want, and these areas should be provided with appropriate shelter, heat and structures for protection and comfort from cold weather and hot sunny days. Regarding plants, depending on the species, they can lose their leaves in autumn/winter, reducing the aerial vegetation cover of the enclosure and eliminating animal escape points, but increasing flooring complexity with the accumulation of the fallen leaves [89]. In rainy seasons, not only the humidity of the environment changes, but also its temperature and vegetation cover, since the number and density of leaves increase [90,91,92]. In addition, hormonal changes can occur due to the initiation of the reproductive season [93]. Thus, complexity through changes in environmental conditions (temperature, light, humidity, etc.) may affect an animal’s experience of complexity, and these influences depend on biological and management traits (i.e., animals housed indoors vs. outdoors, terrestrial or aquatic or amphibious, solitary or social, prey or predator, etc.).

All the cited examples indicate changes in the complexity of the environment seasonally or annually, but diurnal changes can also occur, such as changes in the schedule or in the way animals are offered food (environmental food enrichment schedules [94]); in the dynamics of social relationships, where conflicts can arise due to disputes between individuals [95]; and in temperature, luminosity, and daily humidity due to the day/night cycle, etc.

Daily, seasonal, or annual variations in the complexity of the environment do not mean bad situations for captive animals, as these variations will stimulate different behavioural and physiological states [96,97,98]. However, these changes can mean a decrease in animal welfare if caregivers do not offer the animals the possibility of choice. For example, with the daily or seasonal variation in temperature, if the animals cannot choose between cold and hot environments within their enclosure, due to the lack of installation of heaters, there is a great chance that the animals will experience thermal stress, which will decrease their welfare [99]. One thing we do not know is to what extent animals adapt to local climatic conditions; for example, people living in equatorial zones of the world perceive temperate zones as much colder than local residents [100]. Perhaps equatorial species might perceive long daylengths and significant variation in day length over the year as stressful.

Some zoos are creating large and naturalistic enclosures, such as rainforest enclosures, to provide habitat complexity for many animal species [101]. However, while recreating a rainforest indeed increases complexity, it also largely increases the risk of spreading diseases and losing animals, as this large and complex enclosure may prevent identifying sick animals promptly. Once a sick animal is identified, the next task is catching the animal in the large enclosure for treatment. The challenge for zoo managers is thus the need to create habitat complexity in a way that husbandry practices are not compromised.

Finally, enclosures’ complexity and space have been used as important parameters for welfare evaluation by analyses of a space use index, gaining an inherent value in welfare audits [102]. The use of this index enables a meaningful and repeatable evaluation of the use of different zones within the enclosure, allowing the evaluation of the effect of environmental enrichment on the use of certain less-explored zones of the enclosure [103].

All display zones should have some function and should therefore be used by the inhabitants; if certain zones are avoided by the animals, they may be considered inadequate or of limited value [104]. Although many indexes assess the uniformity of enclosure use as an indicator of suitability [104,105], caretakers must be aware of aspects related not only to the uniform use of the enclosure’s zones, but also to whether the animal is using most of the resources made available to it in each zone. Some exhibit zones may also be of great value to the animals but used only for short periods of time (e.g., defecation area). The comparative value between occupancy zones is not the only parameter indicated for enclosure use evaluations, as animal preference should also be considered in welfare evaluations. Evaluation indexes can help researchers and practitioners to understand animal preferences and develop more biologically relevant enclosures [106].

Although most appropriate when used in conjunction with another assessment tool, enclosure use assessments can be particularly valuable when evaluating the effect of differences in zone use between individual animals [107] or enclosure modification [108]. For example, studies on the antelope sitatunga (*Tragelaphus spekii*) revealed that some resources in the enclosure were accessed differently by individuals of the herd at different ages [106]. A study with flamingos revealed that they use indoor and outdoor enclosures differently at various times of day and night, but this difference could be related to the natural behaviour of the species rather than to some dysfunctional zones of the enclosure [103].

Thus, identifying zones that are avoided or disputed by the animals may allow caretakers to modify the enclosure design by removing components that are actively avoided or by providing components in sufficient numbers to avoid competition [103].

## 2. Why Is It Important to Live in a Complex Habitat?

Attention to the enclosure’s complexity is important because, as stated before, it offers more positive experiences in the four functional domains of welfare (i.e., environment, nutrition, health, behaviour and interactions), favours physiological homeostasis and a good mental state, enhances animal welfare, and improves visitor experiences.

The expression of natural behaviours when in captivity is one of the most used welfare measures [109,110]. An increase in the diversity of behaviours indicative of positive valence states expressed by the captive animals is linked to improved animal welfare [85,111]. Thus, if an increase in the diversity of positive behaviours is observed after the insertion of complexity in an enclosure, then the enclosure renovation can be considered effective [112]. Remember that complexity is dynamic, changing according to temporal and dimensional scales [113,114]. Thus, temporal and spatial variations in behavioural diversity are expected.

Exhibits that provide stimuli to promote the expression of natural behaviours also reduce the expression of unnatural/unwanted behaviours [115]. Unnatural behaviours or abnormal behaviours are those not registered in the wild but registered in captivity or those super-expressed or sub-expressed in the captive environment [116]. Unnatural behaviours are normally linked to low welfare and frustration because animals have the motivation to express some appetitive behaviours but are prevented from expressing appetitive behaviours [117]. However, there is a growing debate about the effectiveness of using natural and unnatural behaviour expression as indicating high or low welfare, since some natural behaviours decrease welfare (e.g., fighting) and some unnatural behaviours increase welfare (e.g., the use of computer games for primates) [118,119]. By offering a complex habitat, with the addition of environmental enrichment, and by applying animal conditioning sessions, the chance of reducing or even eliminating unwanted behaviours increases [28,120,121,122].

Complex enclosure design and the provision of different stimuli gives the animals the opportunity to express appetitive and consummative behaviours, favouring the maintenance of homeostasis [123,124]. Health is favoured when the body functions are normal, allowing animals more positive experiences, which can influence their welfare [78,125]. Additionally, reproduction and production (animal-originated products such as eggs, meat, milk, etc.) are improved in healthy animals [97,126,127,128,129].

Another relevant point to be mentioned is animal genetics. Actions involving biodiversity conservation are focusing on genes as a key element of species adaptability [130]. In recent decades, the development of genetics and genomic approaches have revolutionised conservation biology [131]. The study of genetic diversity is a valuable option to decipher evolutionary changes over time, while epigenetics opens a possibility to assess immediate responses to environmental changes, which potentially can induce spontaneous and random modifications in DNA methylation patterns that can be passed on to future generations [132]. DNA methylations can be used as biomarkers of past and present environmental stress, as well as biomarkers of physiological conditions [131,132,133]. Since an enclosure’s complexity can decrease captive stress [32], it can reduce the deleterious effects of DNA methylation [134], increasing the role of the captive population in conservation efforts [135]. This information shows how the integration of epigenetics (analysis of DNA methylation profiles) and animal husbandry with conservation biology can corroborate data on the physiological, biological, and ecological status of animals [131,136].

Once science has proven the influence of environmental management on the genetics of animals, the institutions that keep fauna in ex situ conditions have a double responsibility and commitment to the welfare protocols. Modern zoos and aquariums, as well as scientific and conservationist breeding facilities, have an ethical responsibility to provide the best care for the animals and to increase the comprehension of the species’ biology, applying management techniques that satisfy the physical and psychological needs of the animals [137]. Guaranteeing physical, mental, and emotional health to animals are ways to ensure species fitness, permitting the species to thrive and become capable and viable for population management programs [138]. If animal populations kept under human care can perform genetic exchange with free-living individuals and vice versa, these institutions will be valuably contributing to conservation by the application of the meta-population approach, enabling an increase in genetic variability within the concept of one conservation [138] and the One Plan Approach guidelines [139].

In conclusion, it is important to use science to plan enclosures that are functional and compatible with the species and the number of individuals that inhabit it, favouring their development, learning and interactions with the environment and with other individuals, thus contributing to conservation and education actions. When we use science in favour of animals, the results are also favourable to science, which in turn will benefit animals again with improvements in husbandry and in the physical and mental states (the Five Domains of Welfare) [140,141] (Figure 1).

## 3. How Does Complexity Affects Species?

The complexity of an enclosure may vary according to the species that inhabits it. The same enclosure can have low complexity for a primate species and, at the same time, offer high complexity for a lizard species, for example (Figure 2). The environmental complexity, then, in addition to the variety in the abiotic and biotic components mentioned before, will depend on the animal species and its capacity to perceive it.

Animal species have a set of sensory receptors that allow them to capture, process, and respond to different environmental stimuli [142]. These sensory receptors can be simple or complex, shared or not by different animal groups [143]. Animal species can vary in their sensory capabilities, with some using more visual cues (birds and primates [144,145]), others more auditory cues (amphibians, birds, and bats [146,147,148]), and others more tactile cues (fish and cave living animals [149,150]) from their environment, for example. The combination of using different senses is what allows animals to perceive the complexity of their environment [151]. After the stimuli perceived by sensory organs are processed in the brain, the animals respond appropriately to them [152]. However, brain morphology also varies between species, with the brain being modified in areas intended for processing stimuli captured by sensory organs [153,154]. For example, birds have olfactory bulbs (i.e., site for processing olfactory stimuli) varying from little to highly developed, with more basal birds (Ratites, Anseriformes, Columbiformes, etc.) having less developed bulbs than more derived birds (parrots and songbirds [155]). Reptiles have a well-developed olfactory bulb [156]. Reptiles probably would be benefited more by olfactory complexity in their enclosures than birds, depending on the species. Therefore, the brain capacity to process environmental stimuli is another important characteristic that must be considered when evaluating how animals respond to environmental complexity [50].

Another important consideration when assessing environmental complexity is the animal’s ability to perceive time scales. Small animals with a high metabolic rate perceive more information from the environment per unit of time compared to large animals with a low metabolic rate [10]. For example, flies can perceive more information per unit of time than turtles, perceiving the time scale more slowly [10]. Do animals that perceive time scales more slowly need more complexity in their environment? The temporal scale is involved in the habitat selection of species, since the disproportionate use of certain areas of the environment is linked to the presence of important resources for the animal, and these resources vary in space and time [157]. Resources are environmental conditions that influence biological fitness, such as resting, foraging, breeding, shelter from predators, etc. [158,159]. Therefore, both animals that perceive time more slowly and those that perceive it more quickly will select the habitat according to their needs at that moment, and the variation in possibilities of choice of resources helps the animals to supply their demands, consummating the actions for the which they were motivated, maintaining their homeostasis, and improving their welfare [159]. Thus, environmental complexity must be offered to all animals, regardless of their ability to perceive time scales. Changes in habitat complexity, however, are expected to be perceived differently, with slow time perceivers experiencing variation in habitat complexity less rapidly than fast time perceivers. Logistically, for human caregivers, it is easier to provide complexity to animals that are slow time perceivers due to the fact that humans are slow time perceivers. A recommendation for caregivers would be to evaluate the annual rhythm of each species to try to provide different but important stimuli in each life phase (such as migration, reproduction, hibernation, etc.) [160,161].

Environmental complexity offered to the animals is influenced by human perception of complexity, and the human sensory ability is often limited compared to that of other animals. As examples, humans cannot perceive ultraviolet light [162] and have lower olfactory and auditory capabilities compared to many animal species, depending on their sensitivity to the odorants [148,163,164]. In this way, the environmental complexity offered to animals kept in human care is often based only on human perception and not on the animals’ sensory capacity [165,166,167]. For example, many bird species can detect ultraviolet light and the ability can influence foraging, reproduction, and welfare of the birds [168]. Pekin ducks (*Anas platyrhynchos*) reared under UV lights showed decreased physiological responses of stress [169]. In another study, some individual starlings (*Sturnus vulgaris*) showed preferences for UV light exposition, but with no behavioural changes [170]. Environmental characteristics other than light provided by caregivers can also influence the welfare and behaviour of animals. For example, flamingos are birds that build tall, cone-shaped mud nests [171]. If in a flamingo enclosure a muddy area is not present (no flooring complexity), reproduction will be compromised [172]. This muddy area may be perceived by humans as a dirty area of the enclosure and not be offered to the animals. Consequently, human perception of the complexity of the environment that is different from the animal’s perception can harm the maintenance of the species in human care and decrease its welfare [172]. Therefore, offering adequate environmental complexity to animals stimulates natural behaviours and an adequate physiological functioning of their organisms [173]. Even within species there can be differences in perceptive abilities; for example, some human females are tetrachromats and can perceive millions of more colours than trichromats [174,175].

## 4. Exhibit Renovations, Environmental Enrichment, and Concepts Aiming at Habitat Complexity

The enclosure’s design should stimulate species-specific behaviours. By renovating the animals’ enclosures, managers and caretakers can promote habitat complexity that will offer to the animals different stimuli and the opportunity of choice and control of the environment, which ultimately will enhance their welfare [176,177]. The enclosures must provide necessary features and structures so that animals will display a range of welfare-related behaviours and should also provide positive husbandry practices [178].

However, institutions that do not have funds for a complete or radical renovation of their exhibits must use environmental enrichment. Environmental enrichment is a technique that offers environmental stimulation to animals from the insertion of physical, sensory, food, and cognitive structures in the enclosures, or the insertion of other individuals of the same species or of different species in the enclosures (social stimulation) [33,34]. Artificial enrichment, for example, may provide the necessary environment and it can be considerably more durable, sturdy, and easier to clean, requiring far less maintenance and lower monetary costs [178]. Of course, the use of environmental enrichment is important in its own right because it increases variation inside the exhibits, and should be offered not only in poorly designed exhibits but also in renovated exhibits.

The use of environmental enrichment has already been shown to be efficient in increasing the environmental complexity of enclosures, positively influencing animal welfare. The modification of the enclosure design associated with physical structures that stimulated crawling behaviour instead of jetting swim behaviour decreased the mantle injuries of *Eledone cirrhosa* octopuses, improving their behaviours and physiology [179,180]. Captive Malayan sun bears (*Helarctos malayanus*) housed in enriched enclosures presented better welfare than those housed in barren enclosures, based on behavioural measures [181]. Offering enclosures large enough to allow rectilinear behaviours for snakes can increase their welfare [182]. Chimpanzees used more areas of the enclosure, showing behaviours more similar to those observed in free-living groups after moving to a naturalistic and enriched enclosure [183]. The same was observed when chimpanzees gained more space per individual in enclosures with similar complexity, showing that providing adequate space is also important for animals [184]. A recent literature review showed how environmental enrichment can positively affect the welfare of birds, modifying behaviours and physiological parameters [49]. Thinking about the Five Domains of Welfare [140,141], the use of environmental enrichment provides experiences not only in the domain of behaviour and interactions, but also in the nutritional, environmental, and health functional domains, positively affecting the mental domain and the overall welfare of the animals.

One idea to increase enclosure complexity is the so-called Zoo360 Concept [185]. In this concept, different enclosures are linked through passages (tunnels or elevated ways), allowing the animals to visit these different enclosures, experiencing varying degrees of complexity [186]. For example, if a tiger is kept in an enclosure designed with passages, it can choose to stay in its original enclosure or walk through the passages until it reaches another enclosure that could be built differently, presenting to the tiger different physical structures and micro-habitats. Along the way, the tiger can also experience different views (in terms of landscape), with more or less visitors, and experience different sounds and smells.

The use of dynamic architecture is another way to increase the complexity of an animal enclosure. The dynamic structure is currently applied to humans and consists of the construction of buildings that can change format in time due to the use of dynamic elements [187]. Stimuli from external sources, such as winds, sun rays, and rain, are able to elicit automated responses from the building, allowing the systems to achieve a high performance, increasing its efficiency, sustainability, and deliverability, and finally, comfort for the users [188]. Animal enclosures built considering dynamic architecture could offer to the animals inside considerable change in complexity, but this technique has not yet been applied [189].

A simpler version of the Zoo 360 concept is to provide rotational zoo exhibits (Jon Coe) or even exhibits whose barriers can be moved. Farmers use moveable barriers to ensure their livestock do not overgraze a particular area of land; this same concept could be used to vary the size and type of habitat that animals in captivity have access to.

Complex habitats also avoid the negative impacts of the visitors (the so-called “visitor effect” [190,191]), as they allow the animals on display to hide from people. Captive Edwards’ pheasants (*Lophura edwardsi*), for example, decreased feeding and locomotion behaviours because visitors were acting as threats to the birds [192]. In a review study, the authors demonstrated that the existence of hiding areas in the enclosures can reduce the negative effects of the presence of the public for various animal species [44]. The same was observed for five captive felids, where the species housed in enclosures with hiding places preferred to stay hidden when visitors were present, while those housed in enclosures without hiding places exhibited more abnormal behaviours in the presence of visitors [193]. Diurnal mammals and mammals in closed habitats suffered more from the visitor effect than nocturnal and mammals in open habitats [194].

Complex habitats, however, can also cause problems for institutions that maintain captive animals. Firstly, too much stimulation can stress animals in the same way as too little stimulation [195,196]. Thus, stimulation offers should not pass the healthy stress threshold and caretakers need to be aware of this. Secondly, very complex enclosures with many hiding areas can prevent the visiting public from viewing the animals, frustrating them and generating many complaints [129,130], but this can be addressed by using live-streaming cameras that the public can view images from on their smartphone (e.g., San Diego Zoo (USA); Melbourne Zoo (Australia); Houston Zoo (USA) [197,198,199]. This may be because the public fails to realise how important complexity is for the welfare of the animals. Therefore, environmental education activities with the visitors are important to minimise people’s complaints [200,201]. The enclosure can be designed to meet both animal and visitor needs by offering a variety of hiding options such as vegetation, open dens, and shaded elevated platforms while maintaining the animals in view [55].

Furthermore, it is important to evaluate complexity in the different areas of an enclosure. For example, foraging areas might need high levels of changing complexity to improve welfare, whereas sleeping areas might need low and unvarying levels of complexity, depending on the species. Bottlenose dolphins (*Tursiops truncatus*), for example, had their welfare increased when complex cognitive/foraging enrichment devices were offered, compared to non-cognitive foraging enrichment [202]. Thus, it is important to evaluate which areas of the enclosure will require more complexity and which will not. The best enclosure design will depend on the species and has been debated inside and outside the scientific academy [203].

Finally, the development of exhibits that allow animals to control or select the level of complexity that they wish could help in the enhancement of welfare. For example, climbing structures can have branches that can be unfolded to create complexity or folded to reduce complexity. This might be important for animal rehabilitation, animals coming from barren enclosures, older individuals, or individuals with offspring [85,167,204].

In conclusion, exhibit planning in any animal facility should consider four important aspects. (1) Size: the enclosure needs to offer a space large enough to provide appropriate complexity and stimulation for the species that allow opportunities to experience positive physical and mental states, with elevated welfare and fitness; (2) design: naturalistic elements (i.e., ideally recreate ecosystems) and environmental enrichment can increase the complexity of the habitat; (3) education: the enclosure’s complexity and environmental enrichment activities must be aligned with education purposes so that visitors have a pleasant experience and feel connected to nature and animals; and (4) architecture: dynamic architecture associated to structures inside the exhibits that can be manipulated by the animals to increase or decrease complexity could help in the welfare of animals and visitors. By implementing these aspects in exhibit planning, zoos can provide a better quality of life for their species, full of positive experiences for their animals, visitors, and staff.

## Figures and Tables

**Figure 1 animals-13-01277-f001:**
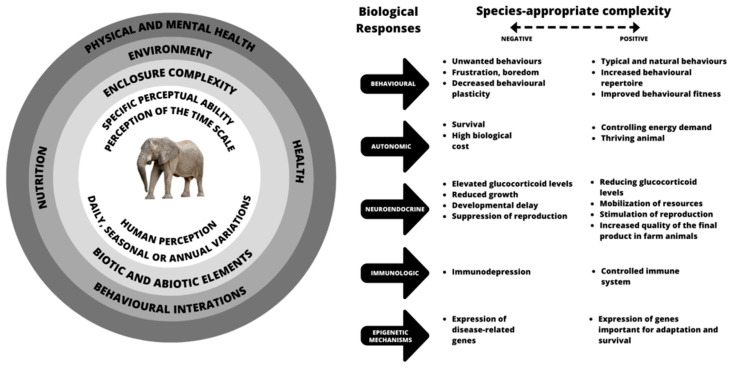
Habitat complexity and the negative and positive consequences on the biological responses of animals kept under human care in the light of the Five Domains of Welfare [140,141].

**Figure 2 animals-13-01277-f002:**
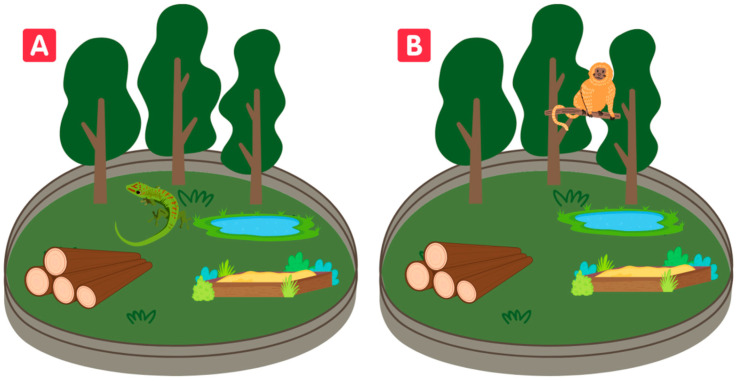
Habitat complexity can be perceived differently depending on the species living in the enclosure. Habitat complexity can be high for a lizard (**A**) and low for a monkey (**B**), even when evaluating the same enclosure.

## Data Availability

Not applicable. No dataset was created.

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
