# Peer review of "Review of the Effects of Enclosure Complexity and Design on the Behaviour and Physiology of Zoo Animals"

_animals, 2023, doi:10.3390/ani13081277_

Round 1

Reviewer 1 Report

This paper describes the importance of environmental complexity to captive animal behavior and welfare. Such information is very complex given the diversity of taxa and environments under human care, and reviews such as this are under published in academic literature. As such, the manuscript would be of value to the academic and professional community.

That said, the manuscript requires significant revision. The concepts are present, however explanations and justifications are applied variably, concepts change between environmental descriptions or affects on animal behavior, and some do not appear to be fully expanded upon and so require further refinement. I included examples and recommendations of areas in needed of further clarification or revision as an aid to the authors in their revision, but therefore are not exhaustive.

Additionally, changes in writing style, grammar, and use of active voice in some but not all sections, I assume individual authors wrote different sections of this manuscript. I recommend further editing to ensure more consistent flow, tone, and voice.

Lns 42-45. There are a lot of concepts within these introductory sentences, as if the reader was jumping into the middle of the manuscript. I recommend reviewing the flow of these sentences and first couple of paragraphs to outline your thesis and concepts. Alternately, could add a couple paragraphs of  Introduction before Section 1.

Ln 49. Is evolution and natural selection an assumption? Consider revising.

Lns 50-52. I think its important to note that these factors are generalizations that inform a species likely experience of environmental complexity, and that they are not mutually exclusive, as many species will fall into extremes of several factors and contradict one another.

Lns 52-93. Given this is a long list that currently blends into the text that follows it, I would recommend considering this being formatted in a was to distinguish it from the main body, either as a table or formatting (e.g., "Species Size: Perception of complexity is a function.... For example, a pygmy marmoset...). This could then highlight the dimensional, temporal, and social factors with biotic and abiotic environments listed above to inform a matrix of considerations.

Ln 61. Would microclimates within the a single latitude fit here?

Lns 61-63. Could also note seasonal climate needs of species in areas outside of their normal range, i.e., providing dark, cool, quiet environments (low complexity) for Holarctic bears to hibernate/brumate in short-day season and their need for high complexity during hyperphagia in longer-day seasons.

Lns 81-82. Biodiversity hotspots are defined by both being high in species endemism, which you acknowledge here, but also in risk of extinction. There are areas of high or higher endemism/diversity which are not "hotspots". Suggest referring to biodiversity, species richness, or similar.  Likewise, should define why diversity increases complexity - through intraspecific interactions, or otherwise?

Lns 83-84. Recommend adding a comparative species, i.e., polar bears with large ranges but more homogeneous habitat vs Rattus spp, or even black bears and a conspecific, whom have smaller ranges but thrive in a larger variety of habitats.

Lns 85-87. Is migration a ecological factor in itself, or an example of a larger concept?  Could this factor be a species trait such as range shifting process which could include migration, emigration, and site fidelity characteristics? 

Lns 92-93. Is brain size and the need or perception of social complexity truly restricted? Meerkats, naked mole rats, schooling fish, and even leaf cutter ants all have complex social dynamics and  interactions that influence their environmental perceptions, complexity, and needs. 

Likewise, I believe a large factor of experiencing environmental complexity is not only the sensory modalities a species has, but its primary means of sensing the world. A Taveta Golden Weaver may perceive its environment similar to a Green Aracari, but a a diurnal Fruit bat species (Pteropodidae spp.) perceives differently from a more nocturnal Leaf-nosed species (Phyllostomidae spp.) given both diet and primary sensory reception (visual vs echolocation).

I mention this as its a critical theme to complexity, which in captivity tends to be anthropocentric, i.e., visual. Species like us which visualize the world will perceptive and have different needs of environmental complexity than those which prioritize sound/echolocation, scent, tactile, electrogenisis, etc as primary sensory receptions, and will surely miss the - potentially more simple - needs of other sensory-able species.

Ln 99. I would recommend including the 'real estate' (space availability and footprint) here, as well. We're often trying to fit animal spaces within existing footprints which surely impedes not only space but complexity. Likewise, guest visibility is often balanced with complexity, either in the form of pathways, experience opportunities (training walls, interactive exhibits and other media) on the guest side with structures that may make animals less visible on the animal's side. Finally, and importantly, I'd recommend noting the complexity of the spaces in which animals their time (exhibits typically and traditionally being more complex compared to holding).

Lns 99-111. I worry two concepts are being blended here: firstly being design of complex spaces and second enriched experience of animals (novelty, temporal predictability, but also behavioral husbandry) which the second is can offset but also be limited by the first. Please consider expanding.

Ln 111. Consider adding, "...create this environmental complexity for species with such biological needs." or similar, to note that the seasonality may not be important to all species cared for in Singapore.

Lns 122-123. the cost may be one consideration, but other non-monetary costs exist in the attempt to replicate or recreate aspects of an ecosystem at a micro-scale, such as loss of management capacity for negative interspecies interactions (aggression, food competition, etc), access to animals to aid in individual welfare (weight management, medication, etc through training), and the assumption that the complexity we create and experience is the same for the individual animals/species inhabiting the space. Please consider including the constraints of such constructed complex environments beyond financial costs.

Lns 130-133. Good information here. This sentence reads awkwardly, please consider revising.

Lns 145-148. Good information provided in this paragraph, but consider defining 'habitat management'. Likewise, the previous text refers to providing complexity and experience through habitat design, but I question if 'meet the needs of the species' is the goal, here.  Later in the paper you describe the Five Domains Model, which describes the emergent experience of the animal based on our management and their environment. This may be a location to inform the reader that designing for complexity (flexible, changeable environments, not static) allows management of animal behavior, choice, and experience beyond basic biological needs. 

Likewise, your example eludes to whole-life planning, which can and should be incorporated into exhibit design and complexity - younger animals generally need more complexity but older animals may need modifications to allow for restricted movement and other reduced abilities. Consider expanding.

Ln 149. Is the objective the complexity of the enclosure here, or the individual? From an enclosure design perceptive, increased social complexity would mean multiple means of feeding opportunities (reduced monopolization), routes of movement through the facility (multiple shift doors, etc), and varied environment in regards to resting, thermoregulation, and other aspects which may spur competition, socialization, and individual comfort.

That said, social complexity as an experience, especially for highly social species, may not depend much on the environment. Data from primates in biomedical research shows that environmental complexity, enrichment, and human-animal interactions are engaging in sterile environments do not compensate for conspecific social interaction. I believe you have the opportunity here to highlight the importance of social engagement as its own type of complexity separate from the environmental experience.  

Lns 156-158. I'm happy to see intraspecific aspects of social complexity, especially polyspecific association here - thank you! The main challenge in captivity is typically health/zoonotic risk, i.e., squirrel and capuchin monkeys, safety risks given constraints of captivity (aggression - I managed a habitat with coatimundi, pudu, and tamarin until we had to remove the latter due to them being too territorial), and that such associations are generally thought to be less mutualistic and more environmental (seasonal aggregation of resources). Again, I recommend noting here the pros and cons. 

Lns. 159-166. Happy to see the Five Domains enter here, recognizing there is a gradient of experience with complexity. I'd recommend noting that not all bad events are negative, assuming they are not chronic, as challenge (through providing events for choice, events for control, and ultimately complexity) leads to an emergent positive mental state. This also provides a good transition into the paragraph below looking at distress and boredom.  

Lns 167-174. I'm not certain if this is the best example for deleterious effects of distress in regards to complexity. Orangutans and other apes have been provided with arboreal access that is underused (see Steve Ross publications) unless there are reasons provided to use the space (food acquisition, thermoregulation, etc). Typically, these aren't the challenges eliciting behavioral pathologies and distress, as they are neural to high on the spectrum of complexity. I'd recommend either discussing how lack of complexity leads to boredom (episodic) and distress (chronic, state), or the positive aspects of complexity either ameliorating behavioral pathologies or enabling animals to be more resilient/antifragle.

Lns 174-194. There is a change in flow with these two paragraphs in that they dont read cohesively but as individual sentences. Consider revising. 

Lns 195-205. This paragraph asks the reader to give a lot of license for generalization. I recommend revising the section on seasonality to less be a description of latitudinal differences in climate and season, and more the complexity through change in environmental conditions (temperature, light, etc) that affect an animal's experience. The reality is there are too many variables in the former (animals housed indoors vs outdoors, terrestrial or aquatic or amphibious, species biological traits and natural history, and general variations across the multitude of taxa managed in zoos) to fit within this manuscript alone. 

Figure 1. I'm not sure the figure is necessary ti illustrate the point being made. Likewise, the caption notes '...can be measured differently..." however the text doesn’t describe a tool for measurement. consider removing or revising.

Lns 216-259. These paragraphs seem to be a expansion of concepts in Section 1. above. Consider incorporating this information above to limit repetition. Likewise, the paragraph beginning below these (starting Ln 260) seems to be a good starting point for this Section.

Lns 231-234. Be careful here, recognizing avian species are derived reptiles, and given the generalizations being made there are a litany of examples to the contrary. Consider being more specific in the example used to limit the reader's distraction to the contrary.

Lns 239-304. The voice, tone, and flow of the text changes at this point, suggesting more editing is needed to refine the manuscript into once voice.

Lns 277-304. Again, and apologies for the recommendations of reorganization, but it seems like these paragraphs are better suited for section 3, below in discussing parameters for measurement of enclosure complexity.

Lns 305-308. Recommend defining social network analysis, pros and cons to comparing captive social experience to wild, why breeding seasonally has temporal variation, etc. Many concepts introduced here that can be expanded on in regards to measurement of complexity in an animal's social and environmental experience.

Lns 309-312. Given the size of most exhibits falls below the resolution for Lidar, and the considerable expense of the technology en leu of information gained, I would question its relevance here. The citations are for wild environments, are there citations on using this technology for captive environments, or similar technology for indoor spaces?

Lns 313-316. In regards to sound complexity, is the best comparable or replicate the wild environment? Sound is dynamic in amplitude, pitch, decibel, etc and animals cue into aspects that are not easy to record, particularly in regards to infrasonic and ultrasonic aspects, depending on their species perceptual needs. Likewise, many species use sound to navigate.

Given the diversity of descriptions possible here and the considerable unknowns, I recommend focusing on measuring an animal's experience with auditory complexity as it relates to their individual and species needs, as authors did above in Lns 239-304.

Lns 324-337. Again, the voice of these two paragraphs differs from those below.

Lns 324-327. More naturalistic enclosures do not necessarily promote complexity or offer more choice and control. Authors qualify this statement with the sentence below, however more justification is necessary. Recommend revising and elaborating on the claim. 

Lns 331-332. Again, I question if environmental enrichment can compensate for poor enclosure design, and if "...funds for renovation..." justify animals living in a less complex environment. Recommend revising.

lns 338-351. This is great justification for enrichment increasing complexity. I would reccomend tying this information to the animals experience (Five Domain Model) vs just increased behavioral opportunities. 

Lns 353-370. These are great concepts - thank you.

Lns 375-401. Love the discussion of pros and cons in complex habitats! thank you

Ln 430-432. Again I’m going to challenge if 'expression of natural behavior' and 'phys homeostasis' is enough. If Five Domains is the model of choice for this manuscript, I encourage using (1) the models language and (2) the emergent properties of experience and mental state.

Ln 436. Increase in the diversity of behavior should be modified to note positive states, as increased complexity could lead to stress and behavioral pathologies which weren't present, and so behavioral diversity increases in relation to  complexity but not as a welfare-friendly outcome.

Lns 467-485. Recommend using this information as an introduction before Section 1 to prime readers for the content of the manuscript.

Figure 2. I don't see where this figure is referred to in the text. Recommend revising or removing.

Author Response

Here you can find a point-by-point response to the editor and reviewers' comments. Please, see the attachment.

Reviewer 2 Report

This is a really nice review covering a broad range of areas relating to housing animals in zoos with suggestions of how practitioners can make the environments more complex and stimulating to encourage natural behaviours and offer choice and control over the environment. There are quite a few older references in here, but I appreciate that the literature isn’t always extensive across this subject area. There are a few more recent references I think you could add, particularly in special issues on this topic across a range of zoo-related journals e.g. JZBG, Animals and JZAR, so I would encourage the authors to add those too where appropriate.

The review would benefit from some minor re-wording to improve readability (as suggested below) and the addition of some sub-headings so it’s clear for the reader how the paper is structured e.g. In section 2 you could use ‘Sensory ability’ etc. This will help the reader to refer back to points of interest.

It is worth considering who the primary target audience is for this paper as it wasn’t clear from the abstract and I think it would be beneficial to mention this. My personal feeling is that it is aimed at zoo practitioners and if so, there are also suggestions below of how to re-frame a couple of potentially inflammatory statements that might result in a negative reaction of the reader, which I know is not the authors intention.

If zoo practitioners are the primary audience it could be worth considering adding sentences about the way in which husbandry routines should be taken into account during exhibit design. Enclosures should be functional, complex and stimulating for the animals (and to some degree the visitors too, as you mention) but there is also a lot of consideration given to the caregivers and how they can conduct husbandry roles (such as cleaning, access for animal care etc) whilst still giving some element of environmental control to the animal. Have you found any references that you could use to include this factor? I feel it’s an important aspect to mention at least and could be included in areas of the review where you mention enrichment and/or food presentation perhaps?

These are some areas that I was a little uncomfortable reading, particularly assuming that zoo practitioners are the primary audience:

Lines 120-123 – There are examples of similar exhibits to the one mentioned at Zurich Zoo which you could mention (in the UK, Monsoon Forest at Chester Zoo springs to mind). If this paper is aimed at zoo practitioners then it’s important not to use language that may be dismissive of those who have taken elements of the ‘ideal’ approaches you mention.

Line 418 – Size of enclosure is a really controversial topic! Large doesn’t always mean best or that the animals will make use it – this is a common point for argument between zoos and animal rights movements so personally I would be cautious about stating this so boldly or you may be cited for and against this argument which misses the overall point of your review article. There are incredibly complex and stimulating environments that may be perceived as small to us but animals use widely and are assessed with high welfare states. It can be far worse to offer a large enclosure that is underused (as you mention earlier in the review, but this can be for other reasons than lack of complexity), providing a negative viewpoint (or lack of visibility to visitors), not to mention the cost of maintaining a large, complex space if not utilised. Optimal size of space is also very subjective and most often not based on any scientific evidence. Zoos cannot replicate the size of wild habitats so how can they define what is the ‘best’ or ‘optimal’ size for an animal? These terms are often used but again can be problematic. If you want to include size here then you could rephrase this in way to describe a space large enough to provide appropriate complexity and stimulation for the species that provides opportunities to perform a range of naturalistic behaviours and maintain fitness.

Suggested text amendments for clarification/ease of reading:

Line 127 – “help animals cope with the constant presence of visitors”. This reads too strong as visitors aren’t always present 24/7 (even during the day). Deleting the word constant would be more appropriate.

Lines 131-132 – “who end up using the most suitable places…” rephrase – do you mean choice over the most suitable area of the environment in any given moment?

Lines 149-151 – sentence needs to be restructured for ease of reading, particularly after “increased if,”.

Lines 157-158 – add the website to the reference list so you can cite it rather than write in full within the text.

Lines 161-166 – sentence restructure for clarity of reading. Suggest “Creating complex enclosures helps to increase the positive experiences of animals across the five domains of animal welfare [60,61], namely environmental, behavioural interactions, health and nutritional domains, with a consequent positive outcome on the mental domain [62,63], provided that exhibits enable individual choice and control over the environment”.

Line 203 – consider adding “state” to the end of the sentence.

Line 239-241 – rephrase first sentence as it is too verbose. Perhaps “Another important consideration when assessing environmental complexity is the animal’s ability to perceive time scales”.

Line 257 – typo, should read “less rapidly”.

Line 258 – typo, should read “complexity to animals…”.

Line 269 – change “deliberated” to “deemed” or ”assumed”.

Line 363 – typo, should read “currently applied”.

Line 406 – delete word “differently”, currently reads “differently different areas of the enclosure”.

Line 410/Ref 165 – what is “the academy”? This needs to be explained in more detail.

Lines 402-410 – The two references don’t really add anything to the opening sentence of this paragraph. You need to explain in more detail why the animals used different areas of the enclosure and if it matches your suggestion that sleeping areas don’t need to be as complex.

Line 417 – restructure first sentence. Suggest “Enclosure planning in any animal facility should consider four important aspects:”.

Line 430 – add “it” before the word “stimulates”.

Lines 431-432 – the term “increases animal welfare” is a little too ambiguous. Do you mean it improves or enhances animal welfare? Use these terms instead if so.

Line 435 – As above – “increase in animal welfare” could be re-written as “increase in positive animal welfare” or “improved animal welfare”.

Line 440 – restructure this sentence for ease of reading. Suggest “Complex enclosure design and the provision of different stimuli enables animals the opportunity to express appetitive and consummative behaviours, favouring the normal body function and maintenance of homeostasis.”

Author Response

(The authors gave the same response as above.)

Round 2

Reviewer 1 Report

The edits and revisions made by the authors have fully satisfied my thoughts and concerns. 
